# (GIGA)byte

DATA RELEASE

# Genome assembly and annotation of the Sharp-nosed Pit Viper *Deinagkistrodon acutus* based on next-generation sequencing data

Xinyu Wang[1,2,3,†], Lirong Liu[2,†], Wenbiao Zhu[2], Shiqing Wang[1,4], Minhui Shi[1,4], Shuhui Yang[3], Haorong Lu[2,*] and Jun Cao[2,*]

1 State Key Laboratory of Agricultural Genomics, Key Laboratory of Genomics, Ministry of Agriculture, BGI Research, Shenzhen 518083, China
2 China National GeneBank, BGI, Shenzhen 518120, China
3 College of Wildlife and Protected Area, Northeast Forestry University, Harbin 150040, China
4 College of Life Sciences, University of Chinese Academy of Sciences, Beijing 100049, China

## ABSTRACT

The study of the currently known >3,000 species of snakes can provide valuable insights into the evolution of their genomes. *Deinagkistrodon acutus*, also known as Sharp-nosed Pit Viper, one hundred-pacer viper or five-pacer viper, is a venomous snake with significant economic, medicinal and scientific importance. Widely distributed in southeastern China and South-East Asia, *D. acutus* has been primarily studied for its venom. Here, we employed next-generation sequencing to assemble and annotate a highly continuous genome of *D. acutus*. The genome size is 1.46 Gb; its scaffold N50 length is 6.21 Mb, the repeat content is 42.81%, and 24,402 functional genes were annotated. This study helps to further understand and utilize *D. acutus* and its venom at the genetic level.

**Subjects** Genetics and Genomics, Animal Genetics, Evolutionary Biology

**Submitted:** 19 May 2023

\* Corresponding authors. E-mail: luhaorong@genomics.cn; caojun1@genomics.cn

† Contributed equally.

Preprint submitted at http://chinaxiv.org/abs/202308.00139

Included in the series: ***Snake Genomes*** (https://doi.org/10.46471/GIGABYTE_SERIES_0004)

## CONTEXT

*Deinagkistrodon acutus* is a species of venomous pit viper, a member of the suborder Ophiopodes and the Viperidae family. It is commonly known as the Sharp-nosed Pit Viper, as well as hundred-pacer viper, five-pacer viper, Chinese moccasin, and Long-nosed Agkistrodon (Figure 1) [1, 2]. Mainly acting in the lungs, *D. acutus* venom is predominantly hemotoxic and can lead to abnormal coagulation and promote tissue damage, edema and acute renal failure, among other reactions [3]. *D. acutus* is widely distributed in southeastern China, Laos and northern Vietnam, and has significant commercial and medicinal value due to its large body size and venom [4, 5]. At present, research is mainly focused on the toxic components of the venom, the analysis of the symptoms of patients bitten by *D. acutus*. Also, its utilization of venom is studied, such as the *in vitro* antibacterial, antithrombotic and anticoagulant activity of specific venom proteins [6–9]. High-quality genomes facilitate the discovery of genes associated with the snake's venom, which in turn can help researchers better understand and utilize the diverse bioactivities of the venom.

Based on next-generation sequencing data, our study assembled and annotated the genome of *D. acutus*. Our research provides essential data support for the discovery and utilization of genes related to snake venoms, and to understand better the phylogeny and evolution of snakes.

**Figure 1.** An individual of *D. acutus* photographed by Diancheng Yang.

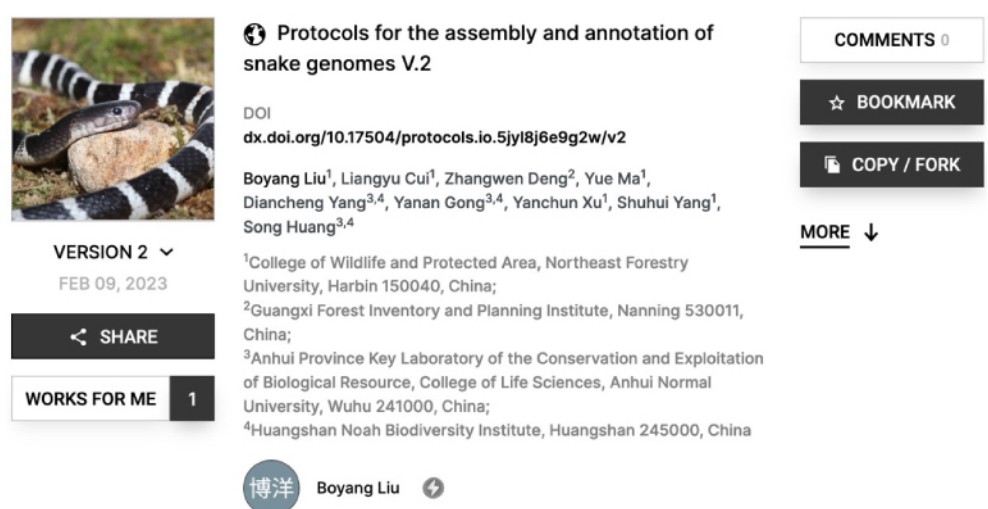

**Figure 2.** Protocol collected from protocols.io for sequencing snake genomes [10]. https://www.protocols.io/widgets/doi?uri=dx.doi.org/10.17504/protocols.io.5jyl8j6e9g2w/v2

## MATERIALS AND METHODS

### Sample collection and sequencing

A specimen of *D. acutus* (NCBI:txid36307) weighing 781 g was obtained from Huangshan City, in Anhui (China), for genome assembly and annotation. The liver, stomach, kidney and muscle tissues were collected for RNA extraction. Additionally, two other muscle tissues were taken for DNA extraction before Whole Genome Sequencing (WGS) and single-tube long fragment read (stLFR) sequencing. We extracted the *D. acutus* DNA, constructed the library and performed paired-end sequencing according to the protocol described by Liu *et al.* (Figure 2) [10]. Sample collection and experimental procedures were approved by the Institutional Review Board of BGI (BGI-IRB E22017).

**Table 1.** Genome assembly data relative to the *D.acutus* genome assembled in this study.

| Item | Category | Size |
|---|---|---|
| | stLFR (Gb) | 164.75 |
| | WGS (Gb) | 96.76 |
| | RNA-seq (Gb) | 10.42 |
| Sequencing data | Assembled genome (Gb) | 1.46 |
| | Longest Contig (Mb) | 0.52 |
| | Contig N50 (Mb) | 0.03 |
| | Longest scaffold (Mb) | 39.38 |
| | Scaffold N50 (Mb) | 6.21 |
| | GC content (%) | 37.9 |

## Genome survey, assembly, annotation and assessment

We used the 25× WGS sequencing data to estimate the size of our assembled *D. acutus* genome. Kmerfreq from GCE (v1.0.2, RRID:SCR_017332) was used for *k*-mer frequency statistics. The output showed that 32,372,553,516 *k*-mer fragments ($k$ = 19) were obtained. Next, these results were input into GCE with the heterozygous mode ($k$-mer depth peak of 21) to evaluate genome size, heterozygosity and other parameters [11].

The stLFR data were used to generate the genome assembly using Supernova (v2.1.1, RRID:SCR_016756). To make the assembled sequences more complete, we used GapCloser (v1.12-r6, RRID:SCR_015026) and the WGS sequencing data to fill gaps. Also, to remove redundant sequences from the genome, we used redundans (v0.14a) [12]. The final genome was obtained using the method described in Figure 2. We used *de novo* prediction and homology-based approaches to identify the repetitive regions in the genome assembly. The homology-based prediction was performed using Blastall (v2.2.26) [12]. Specifically, we mapped the protein sequences from the UniProt database (release-2020_05) of *Pseudonaja textilis*, *Crotalus tigris*, *Thamnophis elegans* and *Notechis scutatus* to the *D. acutus* genome assembly. Annotation and assessment were performed according to the protocol described by Liu *et al.* [10].

To construct a phylogenetic tree, we used OrthoFinder (v2.3.7, RRID:SCR_017118) [13] to search for single-copy orthologs among the protein sequences of *Rana temporaria* (GCA_905171775.1), *Gopherus evgoodei* (GCA_007399415.1), *Podarcis muralis* (GCA_004329235.1), *Thamnophis elegans* (GCA_009769535.1) and *Pseudonaja textilis* (GCA_900518735.1).

## DATA VALIDATION AND QUALITY CONTROL

We used the 164.75 Gb main result file generated by stLFR sequencing to assemble a 1.46 Gb *D. acutus* genome. The genome's longest and N50 scaffolds were 39.38 Mb and 6.21 Mb, respectively (Table 1), indicating that the genome is highly continuous. Comparing the final genome with the 3,354 Benchmarking Universal Single-Copy Orthologs (BUSCOs) from the vertebrate_odb10 database, we found that 87.2% of the 3,354 vertebrate genes (i.e., 2,924 genes) were complete in the *D. acutus* genome; only 245 (7.3%) and 185 (5.5%) genes were BUSCO fragments and deletions, respectively.

In our *D. acutus* genome, the total length of repetitive sequences is 642 Mb, accounting for 42.81% of the genome (Table 2, Figure 3). Based on our *de novo* prediction, we counted the contents of various repetitive sequences. The most dominant repeat elements were long interspersed nuclear elements (LINEs) (443 Mb), followed by long terminal repeats (LTRs) (180 Mb), DNAs (26.43 Mb) and then short interspersed nuclear elements (SINEs) (0.94 Mb). The LINEs and LTRs contents were 29.53% and 11.99%, respectively (Table 3). Repeated

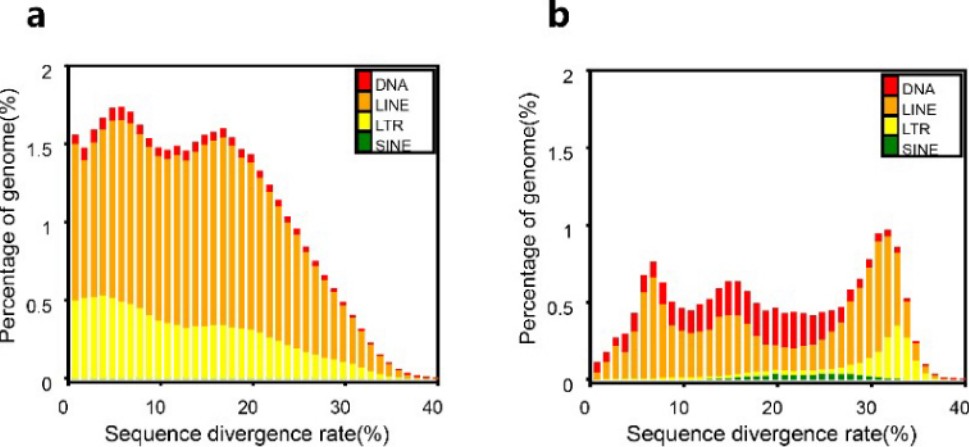

**Figure 3.** Distribution of transposable elements (TEs) in the *D. acutus* genome. The TEs include DNA and RNA transposons (i.e., DNAs, LINEs, LTRs and SINEs). (a) Divergence rate distribution of the *de novo* sequences. (b) Divergence rate distribution of known sequences.

**Table 2.** Statistics for repetitive sequences in the *D. acutus* genome.

| Type | Repeat Size | % of genome |
|---|---|---|
| Trf | 49,665,678 | 3.158437 |
| RepeatMasker (RRID:SCR_012954) | 254,179,490 | 16.16428 |
| Proteinmask | 190,282,517 | 12.100819 |
| De novo | 636,067,480 | 40.45005 |
| Total | 673,253,494 | 42.814856 |

**Table 3.** Statistics for the repetitive sequences (*de novo*) from our *D. acutus* genome.

| Type | Length (Bp) | % in genome |
|---|---|---|
| DNA | 27,712,037 | 1.762318 |
| LINE | 464,343,121 | 29.529418 |
| SINE | 984,426 | 0.062604 |
| LTR | 188,498,215 | 11.987348 |
| Other | 0 | 0 |
| Satellite | 1,180,615 | 0.07508 |
| Simple_repeat | 2,250,205 | 0.143099 |
| Unknown | 2,609,514 | 0.165949 |
| Total | 636,067,480 | 40.45005 |

sequences are important for the self-replication of genetic information, and are closely related to the inheritance and variation of species.

A total of 24,402 functional genes were annotated (Table 4). The results of our gene ontology (GO) enrichment analysis showed that the functional genes of our genome are enriched in biological processes (BP), cellular components (CC) and molecular functions (MF). Among them, cellular process, membrane and binding have the highest content in BP, CC and MF. Our KEGG pathway enrichment analysis using functional genes showed that signal transduction-related genes are crucial in *D. acutus* (Figure 4). Also, the largest number of enriched pathways are related to metabolism. The phylogenetic tree we generated (Figure 5) shows that our data can be used for building species phylogenetic trees. Our tree is consistent with the current knowledge on snake genomes [14]. By



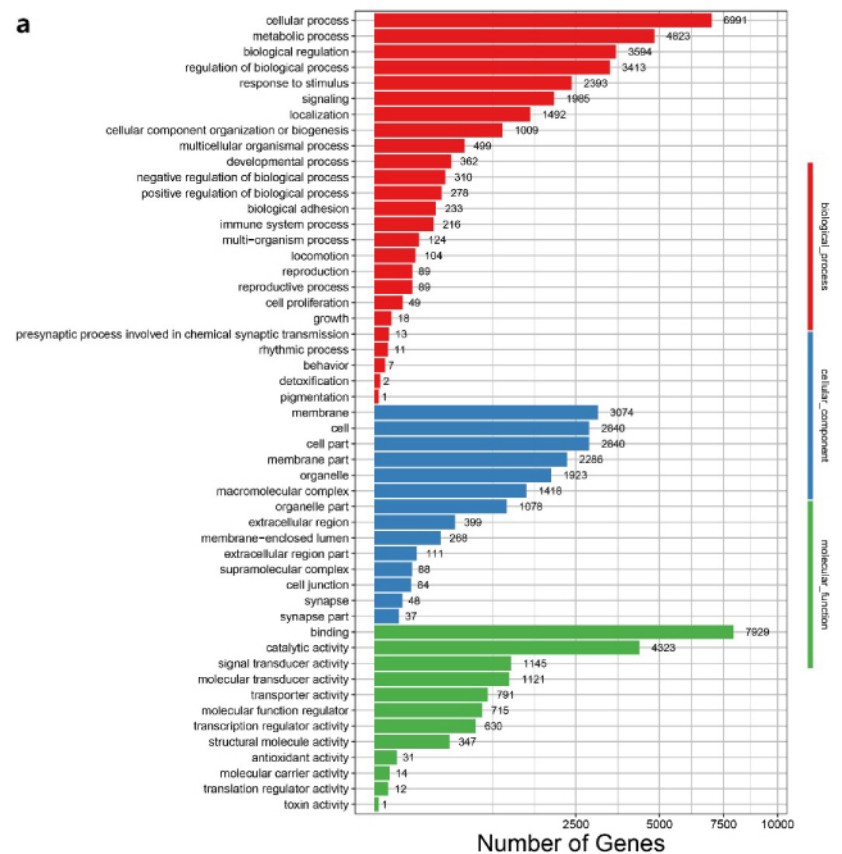

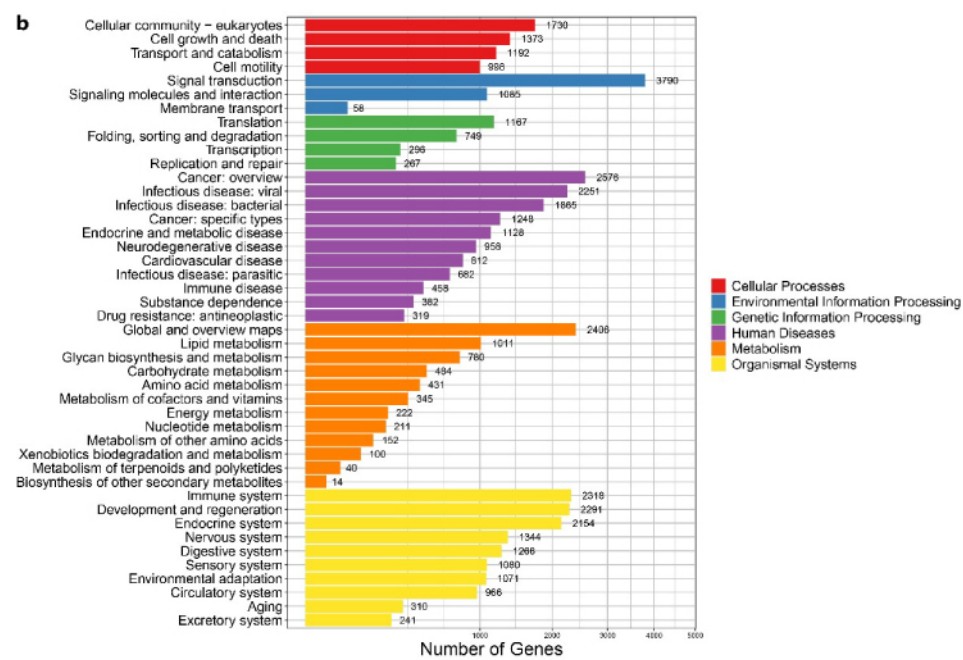

**Figure 4.** Gene annotation of our *D. acutus* genome. (a) GO enrichment. (b) KEGG enrichment.

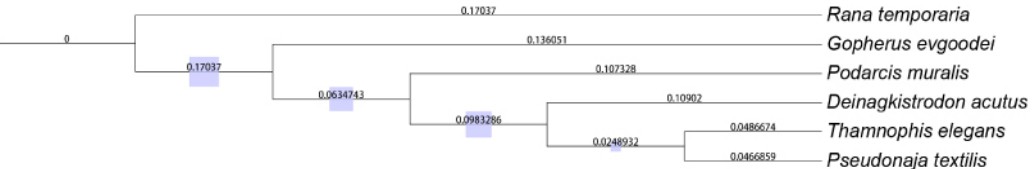

**Figure 5.** Phylogenetic tree reconstructed using single-copy genes from nuclear genomes. The numbers represent the branch lengths. The colored squares represent bootstraps/metadata. The display range is 0.499744 to 1.

**Table 4.** Functional annotation result of our *D. acutus* genome.

|                     | Number | Percentage (%) |
|---------------------|--------|----------------|
| Total               | 24,402 | 100            |
| Swiss-Prot annotated| 19,527 | 80.02          |
| KEGG annotated      | 20,869 | 85.52          |
| TrEMBL annotated    | 22,927 | 93.96          |
| InterPro annotated  | 23,089 | 94.62          |
| GO annotated        | 14,512 | 59.47          |
| Overall             | 23,844 | 97.71          |

comparing our assembled genome data to the chromosome-level genome data of *D. acutus* [1], we demonstrated the successful assembly and annotation of a highly continuous genome of *D. acutus*.

## REUSE POTENTIAL

Our data can be used as a reference genome for others to study *D. acutus*. In addition, it can be used in conjunction with other snake genomes to study the phylogeny and evolution of snakes. Finally, our genome provides data supporting research on snake venom and related toxicology studies.

## CONSENT FOR PUBLICATION

Not applicable.

## DATA AVAILABILITY

The data that support the findings of this study have been deposited into CNGB Sequence Archive (CNSA) [15] of China National GeneBank DataBase (CNGBdb) [16] with accession number CNP0004047. The raw data is also available in SRA via bioproject PRJNA955401. Additional data is available in the GigaDB repository [17].

## LIST OF ABBREVIATIONS

BP, biological process; CC, cellular component; GO, gene ontology; LINE, long interspersed nuclear element; LTR, long terminal repeat; MF, molecular function; SINE, short interspersed nuclear elements; stLFR, single-tube long fragment read; TE, transposable elements; WGS, Whole Genome Sequencing.

## DECLARATIONS

### Ethics approval

Sample collection and experimental procedures were approved by the Institutional Review Board of BGI (BGI-IRB E22017).



## Competing Interests

The authors declare no conflicting financial interests.

## Authors' contribution

JC, HL and LL designed and initiated the project. The snake samples were provided by Anhui Normal University. WZ and SY processed the collected samples. XW, MS and SW performed the DNA extraction and generated the library. XW performed the data analysis and wrote the manuscript. All authors read and approved the final manuscript.

## Funding

Our project was financially supported by the Guangdong Provincial Key Laboratory of Genome Read and Write (grant no. 2017B030301011). This work was also supported by China National GeneBank (CNGB).

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
