## [Editor Report]

Editor’s AssessmentThe sharp-nosed pit viper, Deinagkistrodon acutus is a highly venomous snake distributed across South East and Eastern Asia. To help better understand the evolution of D. acutus, and also provides a molecular basis for the understanding venom production a 1.46Gb in size reference genome was sequenced and described here. This data can be combined with already published and new venomous snake genome data to construct the evolutionary history of venomous snakes and other reptiles. After submission,feedback and improvement to the language, it is now useful to share this data to the community.

---

## [Reviewer Report]

Reviewer name and names of any other individual's who aided in reviewer Inacio Junqueira de AzevedoDo you understand and agree to our policy of having open and named reviews, and having your review included with the published papers. (If no, please inform the editor that you cannot review this manuscript.)YesIs the language of sufficient quality?YesPlease add additional comments on language quality to clarify if needed
Are all data available and do they match the descriptions in the paper? YesAdditional CommentsAre the data and metadata consistent with relevant minimum information or reporting standards? See GigaDB checklists for examples <a href="http://gigadb.org/site/guide" target="_blank">http://gigadb.org/site/guide</a>NoAdditional CommentsI could not find the archive SRR24201538 referring to transcriptomic data in INSDC Sequence Read Archive (SRA).Is the data acquisition clear, complete and methodologically sound?YesAdditional CommentsIs there sufficient detail in the methods and data-processing steps to allow reproduction?NoAdditional CommentsA very essential methodology for a genome sequencing article is the assembly process. The authors mention: "We used stlFR data for genome assembly, to make the assembled sequences more complete" and provide a protocols.io https://dx.doi.org/10.17504/protocols.io.5jyl8j6e9g2w/v2 as the primary methodology reference. However, this protocol does not describe the assembly process step (e.g, software used, parameters tested, outputs, etc). It just reports the initial steps (DNA prep, library prep, and sequencing) and the final bioinformatic step (gene annotation). Please clarify in the main text how the stlFR reads were processed until reaching a draft genome and provide the detailed protocol as a supplementary file or link.Is there sufficient data validation and statistical analyses of data quality? Not my area of expertiseAdditional CommentsIs the validation suitable for this type of data?YesAdditional CommentsIs there sufficient information for others to reuse this dataset or integrate it with other data?YesAdditional CommentsAny Additional Overall Comments to the AuthorRecommendationMinor Revision

---

## [Reviewer Report]

Reviewer name and names of any other individual's who aided in reviewer Jia-Tang LiDo you understand and agree to our policy of having open and named reviews, and having your review included with the published papers. (If no, please inform the editor that you cannot review this manuscript.)YesIs the language of sufficient quality?YesPlease add additional comments on language quality to clarify if needed
Are all data available and do they match the descriptions in the paper? YesAdditional CommentsAre the data and metadata consistent with relevant minimum information or reporting standards? See GigaDB checklists for examples <a href="http://gigadb.org/site/guide" target="_blank">http://gigadb.org/site/guide</a>YesAdditional CommentsIs the data acquisition clear, complete and methodologically sound?YesAdditional CommentsIs there sufficient detail in the methods and data-processing steps to allow reproduction?YesAdditional CommentsIs there sufficient data validation and statistical analyses of data quality? YesAdditional CommentsIs the validation suitable for this type of data?YesAdditional CommentsIs there sufficient information for others to reuse this dataset or integrate it with other data?YesAdditional CommentsAny Additional Overall Comments to the AuthorIn this manuscript, the authors have sequenced and assembled the genome of an important venomous snake. This provides significant data to support future studies on snake evolution. While the content of the manuscript is insightful, there are certain areas where the language could be improved. For example, the title of Figure 2 appears to be incorrectly expressed. I suggest that the author perform a thorough check of the entire paper and make necessary corrections. Furthermore, in Figure 5, the authors need to explain the meaning of the numbers and dots on the phylogenetic tree. The formatting of references, including Latin names, also needs to be checked and corrected, where necessary, to ensure that they are properly italicized. Besides, it is worth noting that the first genome of the sharp-nosed pit viper has already been published (Yin et al., 2016). Therefore, I recommend that the authors include a comparison with older versions of the genome in this manuscript.RecommendationMinor Revision